# `GraphFlow`: A Graph-Based Workflow Management for Efficient LLM-Agent Serving

**Ao Li**[1]  **Shangpeng Yang**[1]  **Fahao Chen**[2]  **Tianheng Xu**[3]  **Peng Li**[1]  **Zhou Su**[1]

## Abstract

Large Language Model (LLM)-based agents demonstrate strong reasoning and execution capabilities on complex tasks when guided by structured instructions, commonly referred to as workflows. However, existing workflow-assisted agent serving systems typically rely on predefined templates and shallow matching mechanisms, which limit their ability to capture deep semantic relationships and generalize to previously unseen tasks. To address these limitations, we propose a new workflow management paradigm that represents workflows using a unified graph, termed *wGraph*, where each node corresponds to an atomic operation. *wGraph* serves as a shared substrate from which task-specific workflows are dynamically instantiated. Building on *wGraph* primitives, we introduce `GraphFlow`, a system that efficiently integrates workflows into agent serving through two key designs. First, adaptive workflow generation dynamically constructs workflows from *wGraph* based on task semantics and constraint requirements. Second, workflow state management exploits *wGraph* structure to efficiently manage Key-Value (KV) caches, reducing redundant computation during agent serving. Extensive experiments across five benchmark datasets show that `GraphFlow` consistently outperforms state-of-the-art methods, yielding an average performance improvement of approximately 4.95 percentage points, while achieving an approximately 4× reduction in memory footprint.

---

[1]School of Cyber Science and Engineering, Xi'an Jiaotong University, Xi'an, China [2]School of Artificial Intelligence, Shandong University, Jinan, China [3]Shanghai Advanced Research Institute, Chinese Academy of Sciences, Shanghai, China. Correspondence to: Peng Li <pengli@xjtu.edu.cn>.

*Proceedings of the 43rd International Conference on Machine Learning*, Seoul, South Korea. PMLR 306, 2026. Copyright 2026 by the author(s).

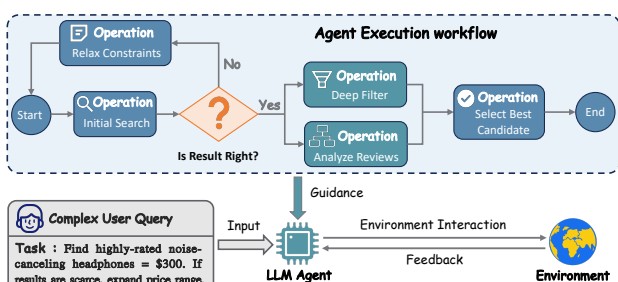

*Figure 1.* Structured agentic workflow for complex online shopping. The agent executes a set of atomic operations (e.g., search, filter, review) to fulfill the user query.

## 1. Introduction

Large Language Model (LLM)-based agents (Wang et al., 2025a; Shang et al., 2025) have demonstrated strong potential for complex task execution, including multi-step planning (Fourney et al., 2024), sophisticated tool orchestration (Zhang et al., 2025b), and multimodal interaction (Wang et al., 2024). By iteratively reasoning and interacting with external environments, these agents can transform high-level instructions into executable actions and gradually complete long-horizon tasks (Yao et al., 2023b). In this process, workflows play a critical role in providing stable and structured procedural guidance (Wang et al., 2025b; Xu et al., 2024; Li et al., 2024). A workflow is a structured composition of operations executed under predefined ordering and control rules, and is widely used to specify the execution logic of complex processes (Yao et al., 2023b; Hong et al., 2024; Wang et al., 2025b), as illustrated in Figure 1.

To introduce explicit procedural structure into LLM-based agent systems, recent approaches augment agents with workflow-assisted execution frameworks (Hong et al., 2024; Qiao et al., 2023; Wang et al., 2025b). These systems typically maintain a repository of predefined workflows (Hong et al., 2024; Josifoski et al., 2023; Qiao et al., 2023), where each workflow encodes the procedural template for a class of tasks. Given a new request, the agent retrieves a candidate workflow by matching the task description against workflow descriptions and executes the selected plan (Xiao et al., 2024; Shen et al., 2025; Fan et al., 2025).

Despite their success, existing approaches suffer from two fundamental limitations. First, existing workflow-based agents rely predominantly on static or retrieval-based workflow construction, which limits flexibility and generalization. Most current systems either execute manually specified workflows (Hong et al., 2024; Yao et al., 2023b; Li et al., 2025) or retrieve a predefined workflow based on semantic similarity between the task and stored workflow descriptions (Wang et al., 2023a; Zhu et al., 2023; Wang et al., 2023b; Qiao et al., 2023). In these approaches, workflows are treated as coarse-grained templates selected from a fixed repository, and the internal structure of workflows is not explicitly modeled or optimized. As a result, they struggle to capture fine-grained correspondence between task requirements and procedural structure, often producing suboptimal execution plans and failing to systematically generalize to unseen or compositional tasks.

Second, existing workflow-based agent systems incur substantial memory redundancy in workflow state management. During inference, modern LLM relies on Key-Value (KV) caches to store intermediate attention states, which are critical for avoiding recomputation and enabling efficient long-context inference (Kwon et al., 2023; Pope et al., 2023). Most workflow-assisted systems manage KV caches at the granularity of individual workflow. However, workflows in practice exhibit extensive operation-level overlap: different workflows repeatedly invoke the same operations, such as identical tool calls, reasoning steps, or verification modules (Khattab et al., 2024; Shang et al., 2025). Under the default per-workflow caching paradigm, these shared operations are associated with multiple separately stored KV states across workflows, resulting in pervasive memory duplication and directly limiting scalability.

To overcome these challenges, we propose GraphFlow, a graph-based workflow management framework for LLM agent systems. At the core of GraphFlow is a global operation graph, termed the *wGraph*, which unifies atomic operations and their dependency relations from multiple workflows into a single structured representation. The *wGraph* explicitly captures operation-level sharing and valid execution dependencies, providing a shared substrate from which task-specific workflows can be dynamically constructed. Building on the *wGraph*, we introduce two core techniques. First, we develop a task-adaptive workflow generation mechanism based on a Graph Neural Network (GNN), which synthesizes new workflows from the *wGraph*. Rather than retrieving a fixed template, GraphFlow learns a topology-aware generation model over the *wGraph*. Given a user query, the model performs inference to produce a task-specific subgraph that is instantiated as the executable workflow. This enables fine-grained recomposition of operations, and allows GraphFlow to construct workflows that adapt to diverse and previously unseen tasks.

Second, we design a topology-aware state management mechanism over the *wGraph* to eliminate redundant workflow state storage. Each operation node may appear in multiple workflows under different execution contexts. To support this shared yet context-dependent execution, GraphFlow decomposes the KV state associated with each operation into a context-independent base and sparse, topology-aware residuals that capture only the differences induced by distinct workflow prefixes. In addition, a path-pruning strategy removes invalid or unreachable transitions in the *wGraph*. This prevents state space explosion and ensures that memory consumption scales only with valid execution trajectories.

We evaluate GraphFlow on five diverse benchmarks spanning mathematical reasoning, complex question answering, and code generation, using multiple backbone LLMs for agent implementation. Across all experimental settings and model backbones, GraphFlow consistently improves agent execution quality by an average of 4.95 percentage points over existing baselines, while reducing the KV cache memory footprint by approximately $4\times$.

## 2. Related Work

**LLM Agents.** Large Language Model (LLM) agents have progressed from pure text generation toward autonomous, goal-driven execution through iterative reasoning and interaction with external environments (Yao et al., 2023b; Park et al., 2023). Foundational reasoning paradigms such as Chain-of-Thought (CoT) (Wei et al., 2022) and Tree-of-Thoughts (ToT) (Yao et al., 2023a) enable agents to explore alternative reasoning paths. To more tightly couple reasoning and action, ReAct (Yao et al., 2023b) and LATS (Zhou et al., 2023) incorporate search procedures and environment feedback to refine execution trajectories. To improve robustness in long-horizon tasks, recent work introduces experience-based reflection and memory mechanisms: Reflexion (Shinn et al., 2023) and ExpeL (Zhao et al., 2024) enable verbal self-correction from past failures, while MemGPT (Packer et al., 2023) and Ghost-in-the-Matrix (Zhu et al., 2023) organize context window as hierarchical memory structures for long-term interaction.

**Agentic Workflows.** Structured workflows are widely adopted to coordinate complex industrial and research processes by organizing multiple operations or agent roles (Li et al., 2023; Hong et al., 2024; Wu et al., 2024; Li et al., 2025; Qiao et al., 2025). Role-based systems such as MetaGPT (Hong et al., 2024) and ChatDev (Qian et al., 2024) encode domain knowledge into predefined Standard Operating Procedures (SOPs). To move beyond rigid templates, subsequent work formulates workflows as programmatic or graph-structured entities to enable modular tool orchestration and iterative refinement, as exemplified by

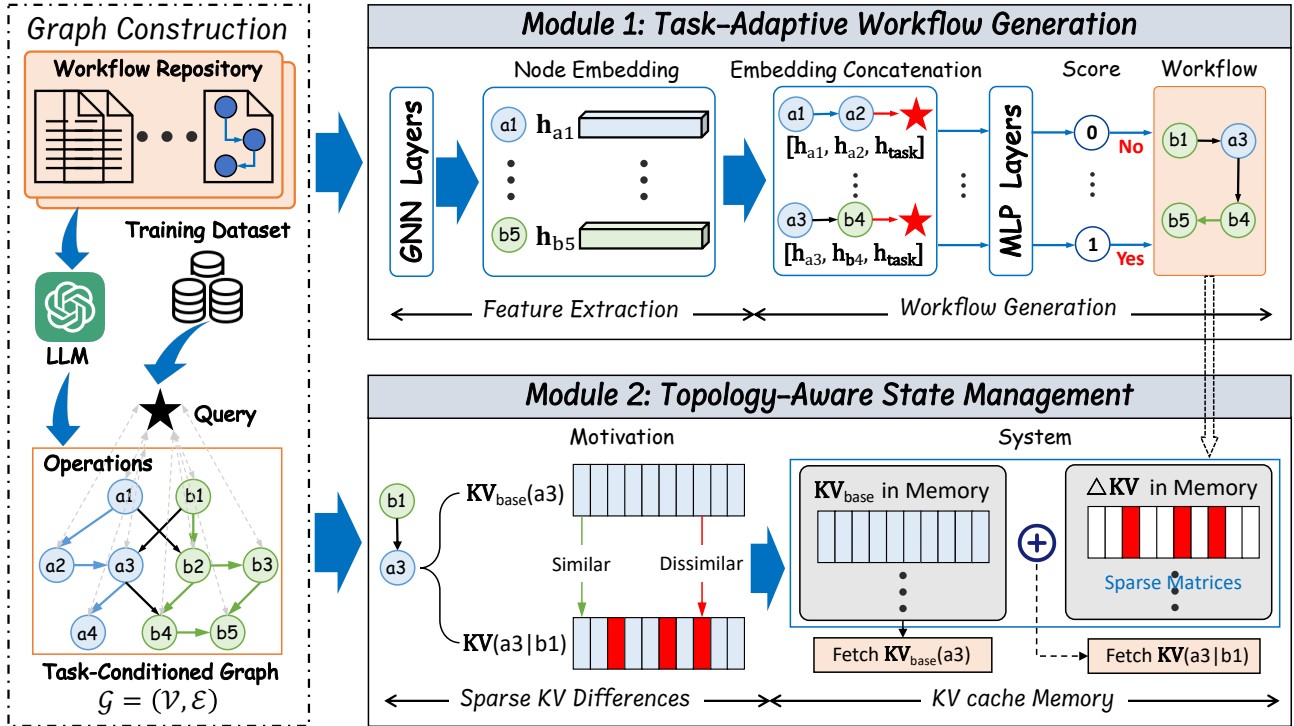

*Figure 2.* The overall framework of our proposed `GraphFlow`.

TaskWeaver (Qiao et al., 2023), LLM-Compiler (Kim et al., 2024), and AFlow (Zhang et al., 2025a). Despite these advances, workflow-assisted serving systems continue to face scalability challenges. Retrieval-centric approaches such as Voyager (Wang et al., 2023a) and Skill-Search (Wang et al., 2023b) retrieve predefined plans as independent units and fail to leverage operation-level structural overlap. From a systems perspective, memory-efficient LLM serving techniques such as PagedAttention and vLLM (Kwon et al., 2023) reduce KV cache fragmentation for general inference, but existing agentic systems still maintain separate KV states per workflow instance. Consequently, substantial memory redundancy arises when workflows share common execution prefixes, a limitation also highlighted in recent studies on prompt sharing and prefix caching (Gim et al., 2024; Zheng et al., 2024; Yao et al., 2025).

## 3. Overview

Figure 2 presents an overview of `GraphFlow`, a graph-based framework for efficient and accurate workflow generation to better support LLM agent execution. `GraphFlow`'s key abstraction is a global operation graph, termed the *wGraph*, which consolidates operations from a workflow repository into a single directed acyclic graph (DAG). Formally, the *wGraph* is defined as $\mathcal{G}_{op} = (\mathcal{V}_{op}, \mathcal{E}_{op})$, where each node $v_i \in \mathcal{V}_{op}$ denotes an atomic operation (e.g., a tool invocation, reasoning step, or verification module), and each

directed edge $(v_i, v_j) \in \mathcal{E}_{op}$ encodes a valid structural or functional dependency between operations. Unlike prior approaches that treat workflows as isolated templates, the *wGraph* explicitly captures operation-level sharing across workflows and serves as a shared substrate from which task-specific workflows are instantiated as subgraphs. Based on the *wGraph*, `GraphFlow` synthesizes workflows on demand by selecting and composing operation nodes along valid execution paths conditioned on incoming queries. This unified representation enables both flexible recomposition of operations and topology-aware state reuse, forming the foundation of `GraphFlow`'s serving pipeline.

`GraphFlow` comprises two main components. The first component is *task-adaptive workflow generation*, responsible for generating a task-specific workflow subgraph from the *wGraph*, given a user query. The second component, *topology-aware state management*, exploits the shared graph structure to manage execution states and enable efficient, correct reuse of KV caches across overlapping operations. We detail these two components in Sections 4 and 5, respectively.

`GraphFlow` operates in two phases. In the *offline preparation phase*, the system constructs the *wGraph*, initializes operation representations, and trains the workflow generation model. Meanwhile, context-free base KV states associated with operation nodes are prepared to support efficient execution. In the *online serving phase*, for each incoming request,

GraphFlow generates a task-specific workflow as a sub-graph of the *wGraph*, selectively reconstructs the required KV states, and sends the resulting workflow to downstream agents for efficient inference.

# 4. Task-Adaptive Workflow Generation

Given the *wGraph* $\mathcal{G}_{op} = (\mathcal{V}_{op}, \mathcal{E}_{op})$ introduced in Section 3, the goal of workflow generation is to construct a task-adaptive workflow that maximizes downstream agent performance. A workflow $\mathcal{W}$ is defined as a directed acyclic subgraph of $\mathcal{G}_{op}$, consisting of a selected subset of operations and their induced dependency relations. Given a task $S$, we seek to generate an optimal workflow:

$$\mathcal{W}^* = \arg \max_{\mathcal{W} \subseteq \mathcal{G}_{op}} \mathbb{E}[f(S, \mathcal{W})], \qquad (1)$$

where $f(\cdot)$ denotes a task-level performance metric (e.g., execution success rate or answer accuracy). Unlike retrieval-based approaches, GraphFlow treats workflow synthesis as a conditional subgraph generation problem, jointly predicting which operations to invoke and how they should be composed.

## 4.1. Task-Conditioned Graph Construction

To condition workflow generation on task semantics, we augment the *wGraph* with a *virtual task node* that injects task-specific information into the operation graph. Given the *wGraph* with operation nodes $\mathcal{V}_{op}$ and edges $\mathcal{E}_{op}$, we construct a task-conditioned graph $\mathcal{G} = (\mathcal{V}, \mathcal{E})$, where the node set is defined as

$$\mathcal{V} = \mathcal{V}_{op} \cup \{v_{task}\}, \qquad (2)$$

and $v_{task}$ denotes a virtual task node representing the input task. To enable global information exchange between the query and all operations, we add bidirectional edges between $v_{task}$ and each operation node:

$$\mathcal{E} = \mathcal{E}_{op} \cup \{(v_{task}, v_i), (v_i, v_{task}) \mid v_i \in \mathcal{V}_{op}\}. \qquad (3)$$

Each operation node $v_i \in \mathcal{V}_{op}$ is initialized with a $D$-dimension feature vector $\mathbf{x}_i \in \mathbb{R}^D$, which encodes its functional semantics, linguistic trigger patterns, and internal execution schemas. The task node $v_{task}$ is initialized with a task feature $\mathbf{x}_{task} \in \mathbb{R}^D$ derived from the input task. The detailed feature construction is described in Appendix B.1. The resulting task-conditioned graph $\mathcal{G}$ can be also represented by a node feature matrix $\mathbf{X} \in \mathbb{R}^{|\mathcal{V}| \times D}$ and a binary adjacency matrix $\mathbf{A} \in \{0, 1\}^{|\mathcal{V}| \times |\mathcal{V}|}$.

## 4.2. Workflow Construction Model

Given the task-conditioned graph $\mathcal{G}$ with node features $\mathbf{X}$ and adjacency matrix $\mathbf{A}$, GraphFlow employs a workflow construction model to instantiate a task-specific workflow. Formally, the constructed workflow is defined as

$$\mathcal{W}_c = \mathcal{M}(\mathcal{G}, \mathbf{X}, \mathbf{A}), \qquad (4)$$

where $\mathcal{M}(\cdot)$ denotes the workflow construction model and $\mathcal{W}_c$ is a connected subgraph of $\mathcal{G}$ representing the instantiated workflow. As shown in Figure 2, GraphFlow's workflow generation consists of two phases: (1) GNN-based representation learning to encode task- and structure-aware node embeddings, and (2) Multi-Layer Perceptron (MLP)-based workflow instantiation to select and compose operation nodes into a valid workflow subgraph.

**GNN-based representation learning.** GraphFlow first applies a GNN to jointly encode latent information of operation node features, task node features, and their structural dependencies. Given the input feature matrix $\mathbf{X}$ and adjacency matrix $\mathbf{A}$, node representations are computed as:

$$\mathbf{H} = \text{GNN}(\mathbf{X}, \mathbf{A} | \Theta_{\text{GNN}}), \qquad (5)$$

where $\mathbf{H} \in \mathbb{R}^{(|\mathcal{V}|) \times D}$ denotes the learned node embeddings, $D$ is the embedding dimension, and $\Theta_{\text{GNN}}$ are the trainable parameters of the GNN. Each embedding $\mathbf{h}_i \in \mathbf{H}$ captures both the intrinsic functionality of operation $v_i$ and its relevance to the input task through message passing with the task node.

**MLP-based workflow instantiation.** Based on the learned node embeddings, GraphFlow constructs a workflow by selecting operation dependencies that are most relevant to the input task. For each candidate operation edge $(v_i, v_j)$ permitted by the underlying *wGraph*, we compute a task-aware compatibility score with an MLP as:

$$s_{i,j} = \text{MLP}\left(\text{CONCAT}[\mathbf{h}_i, \mathbf{h}_j, \mathbf{h}_{task}] | \Theta_{\text{MLP}}\right), \qquad (6)$$

where $\mathbf{h}_{task}$ is the embedding of the virtual task node and $\Theta_{\text{MLP}}$ denotes the parameters of the MLP. CONCAT is the vector concatenation operation. The score $s_{i,j} \in [0, 1]$ measures the likelihood that the dependency from operation $v_i$ to $v_j$ should be included in the workflow for the given task.

Starting from the task node, GraphFlow progressively selects operation dependencies with the highest compatibility scores while enforcing structural validity constraints, until a connected execution subgraph $\mathcal{W}_c$ is formed. This procedure yields a task-specific workflow that balances task relevance and execution feasibility.

# 5. Topology-Aware State Management

Beyond generating workflows, GraphFlow must also support efficient execution. In LLM-based agents, execution efficiency is largely determined by how KV caches are managed. KV caches store the intermediate attention states

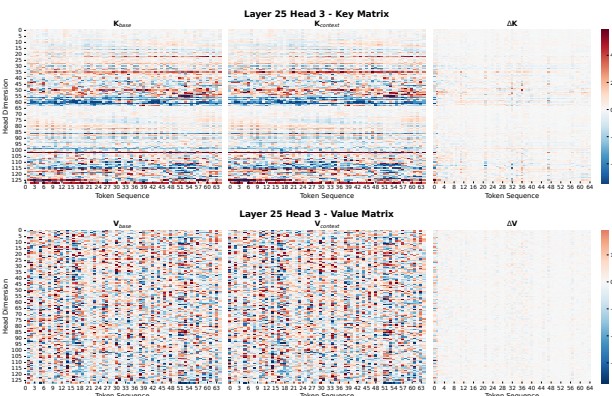

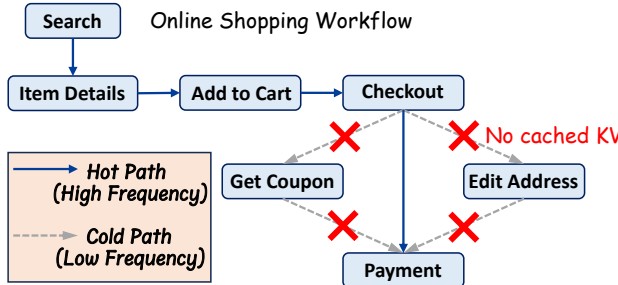

*Figure 4.* Example of effective path pruning.

*Figure 3.* Visualization of KV sparsity. We evaluate the element-wise difference matrices ($\Delta K$, $\Delta V$) between stateful and stateless base KV caches.

produced when processing previous tokens. They are essential for long-horizon and multi-step workflows, as they avoid recomputing past context and directly determine both latency and memory cost during serving.

In `GraphFlow`, each operation corresponds to a fixed textual description (e.g., a tool call, a reasoning step, or a verification prompt). Instead of repeatedly re-encoding these texts at runtime, we associate each operation node in the *wGraph* with its precomputed KV states. Once a workflow is generated as a subgraph, the system can directly assemble the required KV caches from the selected operation nodes, enabling fast execution without repeated prefill.

However, KV management becomes challenging in a graph-based workflow system. A naive design is to cache KV states independently for each operation, computed only from its standalone description. This stateless caching is memory efficient, but it breaks execution correctness. In real workflows, the KV state of an operation depends on its prefix. Ignoring this dependency disrupts attention context and leads to degraded agent performance. At the other extreme, one can store a separate KV cache for each operation under every possible prefix. This fully stateful caching preserves correctness, but it is not scalable. Since operations are shared across many workflows, a single node may appear under massive distinct prefixes. The number of prefix-conditioned KV states therefore grows rapidly with the branching of *wGraph*, causing prohibitive memory overhead.

### 5.1. Differential-based KV Cache

`GraphFlow` addresses this challenge with a differential KV cache design. Our key observation is that, for the same operation, KV caches computed under different prefixes are often very similar. Most entries remain unchanged, and only a small subset is affected by the prefix. We validate this empirically by computing KV caches of the same operation

under different execution prefixes. For a set of representative operations, we compare KV states computed in isolation and with different preceding operations, and measure element-wise differences across layers and attention heads. Figure 3 visualizes these differences as heatmaps and distributions, showing that most entries remain close to zero. Quantitatively, more than 75% of key entries and over 70% of value entries fall below a small threshold, indicating that prefix-induced KV changes are highly sparse. Additional experimental details and extended results are provided in Appendix E.

Based on this observation, `GraphFlow` represents the KV state of each operation in two components: a *base KV cache*, computed once from the standalone operation description, and a *prefix-specific residual*, which stores only the sparse differences introduced by a particular execution prefix. During execution, `GraphFlow` first loads the base KV cache of an operation node and then applies the corresponding residual associated with the current prefix to reconstruct the full context-aware KV state. Specifically, for an operation $v$ under a prefix path $\mathcal{P}$, the KV state is reconstructed as

$$\mathbf{KV}(\mathcal{P}, v) = \mathbf{KV}_{\text{base}}(v) + \Delta\mathbf{KV}(\mathcal{P}, v), \qquad (7)$$

where $\mathbf{KV}_{\text{base}}(v)$ denotes the base KV cache computed from the standalone operation description, and $\Delta\mathbf{KV}(\mathcal{P}, v)$ is a sparse residual that captures only the prefix-induced differences. This design preserves execution correctness while avoiding repeated storage of nearly identical KV tensors.

### 5.2. Effective Path Pruning

Even with differential storage, maintaining residuals for all possible paths in the *wGraph* is unnecessary. In practice, only a small subset of transitions are frequently exercised, while many paths are possible, but rarely or never used. As an example shown in Figure 4, in an online shopping workflow, it is unlikely to "edit address" after "checkout" and the corresponding paths can be pruned to save storage.

`GraphFlow` therefore applies effective path pruning. Using execution statistics, we identify high-frequency transitions and materialize their residual KV states. For rare paths,

residuals are not stored. When such a path is encountered, the system falls back to on-the-fly KV computation.

This policy ensures that memory usage scales with the effective working set of workflows, rather than with the full combinatorial complexity of the *wGraph*. It concentrates storage on common execution patterns, while preserving correctness for rare cases.

# 6. Experiments

## 6.1. Experimental Setup

**Model and Dataset.** We evaluate `GraphFlow` using three LLMs as agent execution backbones: Qwen-2.5-7B (Yang et al., 2024), Llama-3.1-8B (Grattafiori et al., 2024), and Gemma-2-9B (Team et al., 2024). Experiments are conducted on five public datasets spanning three representative reasoning-intensive task domains. Specifically, GSM8K (Cobbe et al., 2021) and MATH (Hendrycks et al., 2021) evaluate mathematical reasoning, HumanEval (Chen et al., 2021) and MBPP (Austin et al., 2021) evaluate code generation, and HotpotQA (Yang et al., 2018) evaluates complex question answering.

For GSM8K and MATH, we use accuracy (%) as the primary evaluation metric, while HotpotQA is evaluated using the F1 score. For code generation benchmarks, including HumanEval and MBPP, we adopt the standard pass@1 metric to measure generation accuracy. All metrics follow standard evaluation settings used in (Zhang et al., 2025a). Additionally, to evaluate serving efficiency, we report the *Time* metric, defined as the 90th percentile inference latency (P90). Unlike average latency, P90 captures the tail behavior of the system, providing a more robust measure of user-perceived performance stability in interactive agent serving (Kwon et al., 2023; Pope et al., 2023).

**Baselines.** We compare against representative workflow construction and planning paradigms in recent LLM-based agent systems. (1) *Vanilla* directly invokes the underlying LLM to solve each task without structured workflows or intermediate planning, serving as a non-workflow baseline. (2) *MetaGPT* (Hong et al., 2024) relies on manually specified, role-based standard operating procedures, where execution workflows are statically embedded into agent prompts and reused across tasks. (3) *LLMCompiler* (Kim et al., 2024) constructs task-specific dependency graphs over tool invocations, serving as executable workflows that support parallel and dependency-aware function execution via a compiler-inspired planner. (4) *TaskWeaver* (Qiao et al., 2023) synthesizes workflows by retrieving top-$k$ exemplars and prompting LLMs to adapt or compose them for task execution, without explicitly modeling workflow topology. (5) *AgentKB* (Tang et al., 2025) augments agent execution by retrieving structured external knowledge to guide in-

termediate reasoning steps, serving as a representative of knowledge-augmented planning without explicit workflow graph modeling. (6) *AutoFlow* (Li et al., 2024) automatically induces structured workflows by extracting task-specific patterns and domain knowledge from prior executions. (7) *AFlow* (Zhang et al., 2025a) treats agent execution as a computational graph and applies automated search algorithms to optimize workflow topology.

## 6.2. Overall Performance

Table 1 presents the agent execution quality and latency of different agent systems across three representative LLM backbones. Overall, `GraphFlow` achieves consistently strong performance across all evaluated tasks and models, indicating robust generalization across both task domains and backbone capacities. Across mathematical reasoning, question answering, and code generation benchmarks, `GraphFlow` attains the strongest results on all three backbones. The consistent improvement across different backbone models suggests that the proposed graph-based workflow construction is not tightly coupled to a specific model capacity or architecture.

The benefits of `GraphFlow` are particularly pronounced on tasks that require multi-step reasoning and structured execution. On the HumanEval benchmark, `GraphFlow` achieves a pass@1 accuracy of 86.2% with Qwen-2.5-7B, and similarly strong results are observed across the other backbones. In particular, `GraphFlow` improves HumanEval pass@1 from 78.1% (AFlow) to 86.2% on Qwen-2.5-7B, and from 75.4% to 82.5% on Gemma-2-9B, indicating robust gains on code synthesis where correct tool-free planning and verification are essential. On the MATH dataset, which emphasizes long-horizon mathematical reasoning, `GraphFlow` yields substantial accuracy gains over static and retrieval-based workflow methods, e.g., improving from 72.1% (AFlow) to 76.4% on Qwen-2.5-7B. These results indicate that adaptively synthesizing workflows from *wGraph* is effective in capturing complex dependency structures that are difficult to encode using fixed templates or retrieved workflows.

In addition to accuracy improvements, `GraphFlow` simultaneously demonstrates favorable efficiency characteristics. As shown in Table 1, `GraphFlow` consistently achieves lower end-to-end inference latency than existing agentic baselines across most task and model combinations. For example, on Qwen-2.5-7B, aggregated across the five evaluated datasets, 90% of requests complete within 12.25s under `GraphFlow`, compared to 14.06s for AFlow. This indicates that task-adaptive workflow generation can produce workflows that are not only more effective but also more concise and execution-efficient, leading to reduced inference overhead. Taken together, these results demonstrate that `GraphFlow` provides a favorable accuracy–efficiency

*Table 1.* Comparison of task performance and P90 inference latency between baseline methods and `GraphFlow`. Experiments are conducted using Qwen-2.5-7B, Llama-3.1-8B, and Gemma-2-9B across five benchmarks spanning mathematical reasoning, complex question answering, and code generation.

| Model | Method | GSM8K | | MATH | | HotpotQA | | HumanEval | | MBPP | |
|---|---|---|---|---|---|---|---|---|---|---|---|
| | | Acc. | Time | Acc. | Time | F1 | Time | pass@1 | Time | pass@1 | Time |
| **Qwen-2.5-7B** | Vanilla | 81.5 | 6.28 | 60.4 | 9.63 | 60.7 | 1.29 | 69.2 | 7.23 | 62.5 | 7.09 |
| | MetaGPT (Hong et al., 2024) | 85.3 | 9.89 | 66.6 | 14.52 | 65.7 | 3.28 | 72.1 | 12.20 | 66.7 | 13.21 |
| | LLMCompiler (Kim et al., 2024) | 85.9 | 10.81 | 68.8 | 14.67 | 65.3 | 3.12 | 74.0 | 12.29 | 65.5 | 14.58 |
| | TaskWeaver (Qiao et al., 2023) | 86.6 | 16.84 | 65.5 | 18.26 | 65.6 | 6.40 | 75.3 | 25.76 | 66.1 | 24.67 |
| | AgentKB (Tang et al., 2025) | 88.3 | 12.30 | 68.3 | 15.85 | 67.0 | 3.99 | 76.8 | 20.71 | 67.9 | 26.07 |
| | AutoFlow (Li et al., 2024) | 87.5 | 14.22 | 68.2 | 17.95 | 66.8 | 4.55 | 77.4 | 18.30 | 67.2 | 22.15 |
| | AFlow (Zhang et al., 2025a) | 89.2 | 12.45 | 72.1 | 16.60 | 67.5 | 3.92 | 78.1 | 18.55 | 68.4 | 18.80 |
| | `GraphFlow` (**Ours**) | **92.1** | **11.59** | **76.4** | **15.22** | **70.4** | **2.17** | **86.2** | **15.76** | **74.7** | **16.49** |
| **Llama-3.1-8B** | Vanilla | 78.8 | 4.96 | 36.6 | 12.58 | 56.7 | 1.66 | 66.4 | 8.76 | 59.6 | 10.25 |
| | MetaGPT (Hong et al., 2024) | 82.7 | 8.98 | 42.3 | 14.68 | 61.2 | 2.68 | 70.2 | 13.62 | 62.7 | 17.38 |
| | LLMCompiler (Kim et al., 2024) | 82.2 | 8.92 | 44.7 | 15.05 | 60.8 | 3.25 | 70.5 | 14.52 | 64.1 | 18.15 |
| | TaskWeaver (Qiao et al., 2023) | 82.8 | 12.98 | 45.2 | 20.22 | 61.1 | 7.55 | 69.9 | 24.80 | 63.6 | 28.35 |
| | AgentKB (Tang et al., 2025) | 84.6 | 8.90 | 46.5 | 18.55 | 62.4 | 5.80 | 71.2 | 18.05 | 63.8 | 18.60 |
| | AutoFlow (Li et al., 2024) | 83.5 | 12.15 | 46.8 | 18.10 | 62.8 | 4.88 | 71.0 | 19.55 | 64.5 | 18.90 |
| | AFlow (Zhang et al., 2025a) | 85.4 | 10.60 | 47.5 | 16.40 | 63.2 | 3.75 | 72.2 | 17.10 | 65.9 | 17.45 |
| | `GraphFlow` (**Ours**) | **88.8** | **8.85** | **52.6** | **15.95** | **68.3** | **2.38** | **76.6** | **13.45** | **72.6** | **15.15** |
| **Gemma-2-9B** | Vanilla | 84.4 | 4.74 | 30.6 | 7.99 | 61.2 | 1.51 | 66.8 | 13.78 | 60.5 | 9.78 |
| | MetaGPT (Hong et al., 2024) | 85.4 | 9.81 | 34.6 | 12.87 | 63.7 | 4.50 | 70.7 | 15.69 | 63.4 | 13.61 |
| | LLMCompiler (Kim et al., 2024) | 85.5 | 8.80 | 35.2 | 14.35 | 63.5 | 3.55 | 71.2 | 16.85 | 64.8 | 14.55 |
| | TaskWeaver (Qiao et al., 2023) | 85.2 | 14.15 | 34.8 | 18.50 | 64.2 | 7.85 | 70.5 | 25.25 | 63.2 | 24.75 |
| | AgentKB (Tang et al., 2025) | 85.5 | 9.35 | 37.5 | 15.85 | 64.5 | 4.15 | 73.8 | 18.50 | 65.5 | 15.05 |
| | AutoFlow (Li et al., 2024) | 85.8 | 11.50 | 36.9 | 16.80 | 64.8 | 5.10 | 72.5 | 20.40 | 65.1 | 18.20 |
| | AFlow (Zhang et al., 2025a) | 86.6 | 10.80 | 37.2 | 15.35 | 66.4 | 4.80 | 75.4 | 19.50 | 66.1 | 17.15 |
| | `GraphFlow` (**Ours**) | **89.2** | **8.75** | **42.5** | **13.10** | **69.8** | **4.05** | **82.5** | **16.65** | **72.8** | **15.45** |

trade-off across diverse tasks and backbone models, further highlighting its technical maturity and practicality for workflow-assisted agent serving.

### 6.3. Ablation Study

**Effectiveness of topology-aware state management.** Figure 5 compares the KV-cache memory footprint (bars, left y-axis) and the corresponding task performance (red curve, right y-axis) for three designs: *stateful KV management*, `GraphFlow`, and *stateless KV management*. Note that the performance metric varies by benchmark: we report accuracy for GSM8K/MATH, F1 for HotpotQA, and pass@1 for HumanEval/MBPP.

Across all five benchmarks, `GraphFlow` substantially reduces KV memory relative to *stateful KV management* while largely preserving performance. For example, on GSM8K, the KV footprint drops from roughly 50 GB to about 11 GB, and on the context-heavy HotpotQA task it decreases from around 85 GB to about 25 GB, compared to *stateful KV management*. Despite this compression, `GraphFlow`'s perfor-

mance remains close to the stateful reference across datasets, e.g., on MATH it retains about 52.6% accuracy versus 53.8% under stateful KV management, and similar small gaps are observed on HotpotQA and code benchmarks. These results indicate that `GraphFlow`'s topology-aware state management can remove large amounts of redundant KV state without materially hurting task execution.

In contrast, *stateless KV management* further reduces memory (typically to 8-17 GB on long-context benchmarks), but consistently incurs a larger performance drop, especially on tasks that rely on multi-step reasoning and long-range dependencies. For example, on MATH accuracy falls to about 39.4%, and HotpotQA F1 drops to roughly 58.6%, highlighting that discarding prefix context breaks cross-operation dependencies critical for correct reasoning.

**Memory footprint under different request batch sizes.** Figure 6 illustrates the GPU memory consumption across varying batch sizes. As concurrency increases, *stateful KV management* shows a substantial rise in memory usage, growing from approximately 0.8 GB at batch size 10 to

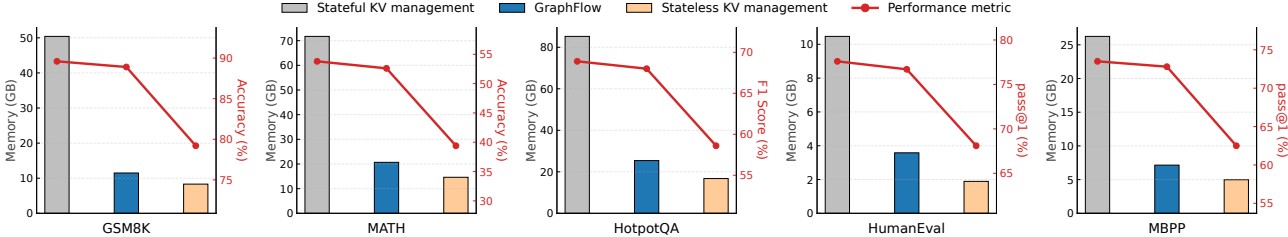

*Figure 5.* Memory-performance trade-off analysis. Comparison of memory footprint and task performance.

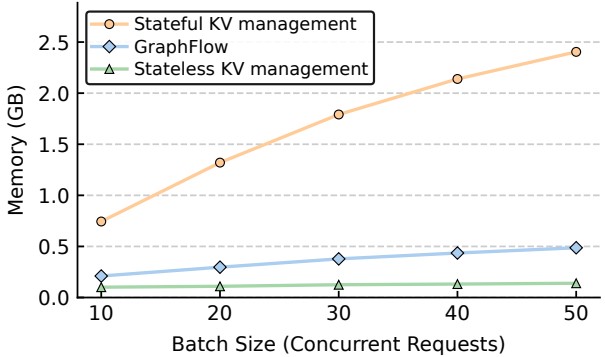

*Figure 6.* Memory consumption with increasing batch sizes.

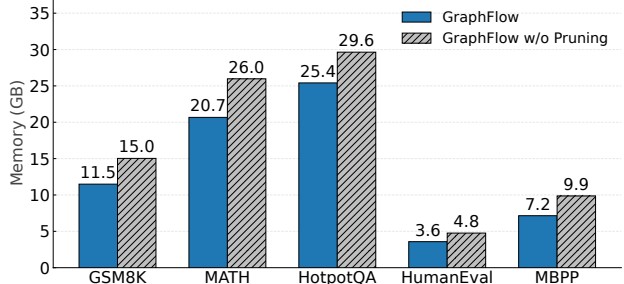

*Figure 7.* Ablation study on path pruning.

over 2.4 GB at batch size 50. This trend occurs because stateful caching maintains independent KV states for each request, causing redundant storage of overlapping operation prefixes. Consequently, memory overhead scales with the number of concurrent workflows, posing a significant bottleneck for high-throughput serving.

In contrast, `GraphFlow` demonstrates substantially improved memory efficiency. Across all batch sizes, `GraphFlow` maintains a compact memory footprint, remaining below 0.5 GB even at the highest concurrency level. While memory usage increases slightly with batch size, the growth rate is significantly slower than that of *stateful KV management*. This behavior indicates that differential-based KV management effectively shares base KV states across concurrent requests and only materializes lightweight prefix-specific differences when necessary, thereby avoiding large-scale duplication of KV states. *Stateless KV management* achieves the lowest memory footprint overall, as it discards prefix-dependent context entirely. However, as shown in prior experiments, this aggressive compression comes at the cost of substantial performance degradation. Overall, these results show that differential-based KV management decouples memory growth from request concurrency, enabling efficient high-concurrency agent serving without incurring the prohibitive memory overhead of stateful KV caching.

**Effectiveness of path pruning.** Figure 7 analyzes the impact of the path pruning module on memory consump-

tion under differential-based KV management. We compare `GraphFlow` with and without path pruning across five benchmarks. Across all tasks, enabling path pruning consistently reduces the KV memory footprint. For example, on GSM8K, memory usage decreases from 15.0 GB to 11.5 GB, and on MBPP from 9.9 GB to 7.2 GB. Similar reductions are observed on MATH and HotpotQA, where pruning removes approximately 5.3 GB and 4.2 GB of KV state, respectively. These results indicate that a substantial fraction of cached KV states correspond to operation transitions never exercised by constructed workflows. The effect of path pruning is particularly pronounced on context-intensive benchmarks such as HotpotQA, where workflows exhibit higher branching factors and longer execution paths. Without pruning, differential-based KV management must retain KV differences for a large set of potential operation transitions. By filtering semantically invalid or unreachable transitions, path pruning restricts KV caching to operation paths reachable during workflow construction, thereby reducing redundant storage.

## 7. Conclusion

We present `GraphFlow`, a topology-aware framework that advances LLM agent orchestration from static workflow retrieval to dynamic subgraph construction over a shared operation graph. By unifying diverse workflows into a global *wGraph*, `GraphFlow` enables flexible procedural composition and explicit reasoning over workflow structure. Experiments across five benchmarks demonstrate consis-

tent improvements in agent execution quality over existing workflow-based baselines. In addition, the proposed differential-based KV cache management reduces memory overhead by approximately $4\times$.

## Acknowledgment

This work was supported by the National Natural Science Foundation of China (No. 62471383).

## Impact Statement

This work aims to advance research on LLM-based agent systems. The proposed framework raises no known ethical concerns in its design, implementation, or data usage. From a societal perspective, `GraphFlow` improves the resource efficiency of agent execution by reducing memory overhead during workflow serving. This efficiency gain can lower the hardware requirements for deploying complex agentic workflows and may help reduce the operational cost and environmental impact of large-scale LLM serving. In addition, by representing agent workflows with explicit graph structures, our approach provides a more structured and transparent execution process, which may facilitate better procedural control and support the development of more reliable agent systems.

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

# A. Experimental and Implementation Details

**Data Preparation and Graph Construction.** To construct the supervision dataset, we leverage GPT-4o to synthesize high-quality execution traces for queries in the training corpus. These traces are parsed to extract atomic operations and their sequential dependencies, forming the ground-truth workflows. The global operation graph $\mathcal{G}_{\text{op}}$ is instantiated by unifying the operation space across all training samples. Specifically, we perform semantic deduplication to merge identical operations into unique canonical nodes, ensuring a compact and consistent action space.

**Model Architecture.** The node feature initialization utilizes the pre-trained all-MiniLM-L6-v2 sentence transformer (Wang et al., 2020), producing initial embeddings of dimension $D = 384$. The structure learning component employs a 2-layer Graph Convolutional Network (GCN) encoder with a hidden dimension of 256 and ReLU activation. The workflow decoding module consists of a MLP head dedicated to edge prediction. This MLP follows a 3-layer architecture with a hidden size of 128.

**Optimization Setup.** We optimize model parameters using AdamW (Loshchilov & Hutter, 2017) with a learning rate of $1 \times 10^{-4}$ and weight decay of $1 \times 10^{-2}$. We train for 20 epochs with a batch size of 64. To avoid enumerating all $N^2$ node pairs, we compute the loss only over candidate dependencies $(v_i, v_j) \in \mathcal{E}_{\text{op}}$ permitted by the *wGraph*. This restriction removes invalid transitions and substantially reduces the effective class imbalance in edge supervision. All experiments are conducted on NVIDIA L20 GPUs using PyTorch 2.1.

# B. Technical Details

## B.1. Feature Initialization Details

In the global operation graph construction phase, we represent each operation node $v_i \in \mathcal{V}_{\text{op}}$ as a focused functional unit extracted from historical workflows. To capture the precise semantics and activation context required for graph-based reasoning, we define each atomic operation as a tuple:

$$v_i = \langle \mathcal{D}_i, \mathcal{P}_i \rangle, \tag{8}$$

where both components describe the intrinsic properties of the node. The two components correspond to the following feature modalities:

1. $\mathcal{D}_i$ (*Semantic Instruction*) encodes the core functional intent of the operation. It is directly derived from the specific task-step description within a workflow, such as *"Translate the story into a linear equation"* or *"Isolate the variable by inverse operations"*. This component provides the primary semantic embedding for the GNN to understand the node's capability.

2. $\mathcal{P}_i$ (*Lexical Trigger Patterns*) captures the explicit linguistic cues that signal the operation's relevance. These are inherited from the workflow's pattern definitions, including strict constraints (*must*: ["solve", "unknown"]) and supportive context (*should*: ["let x"]). These patterns serve as sparse alignment signals to bridge the gap between user queries and atomic graph nodes.

These raw specifications are then mapped into a continuous embedding space of dimension $D$ through a learnable transformation function, $\phi(\cdot)$. This function is implemented as a heterogeneous feature encoder, which generates feature vectors for both operation nodes and the task query. Formally, we have:

$$\mathbf{x}_i = \phi(v_i), \quad \mathbf{x}_{\text{task}} = \phi(\mathcal{Q}). \tag{9}$$

The resulting vectors $\mathbf{x}_i$ and $\mathbf{x}_{\text{task}}$ serve as the initial node features for the GNN encoder, allowing it to capture both the operational semantics and the task-specific context.

## B.2. Network Architecture Details

**GNN Encoder Formulation.** As described in Section 4.2, we adopt a standard GCN (Kipf & Welling, 2016) as the backbone encoder to propagate both structural and task-conditioned information over the task-augmented graph. Given the

adjacency structure defined in Section 4.1, each GCN layer performs localized message passing according to:

$$\mathbf{h}_i^{(l+1)} = \text{ReLU}\left(\sum_{j \in \mathcal{N}(i) \cup \{i\}} \frac{1}{c_{ij}} \mathbf{h}_j^{(l)} \mathbf{W}^{(l)}\right), \tag{10}$$

where $\mathcal{N}(i)$ denotes the neighborhood of node $i$, $c_{ij} = \sqrt{\deg(i)\deg(j)}$ is the symmetric normalization term, and $\mathbf{W}^{(l)}$ is a trainable weight matrix.

By introducing a virtual task node connected to all operation nodes, this message-passing scheme enables task semantics to be globally diffused across the operation graph within a small number of layers. Unless otherwise stated, we use a 2-layer GCN with ReLU activations, following the standard configuration in prior work.

**MLP Decoder Formulation.** Complementing the workflow instantiation process outlined in Section 4.2, this appendix details the specific architecture and optimization strategy of the edge scoring module. The decoder operates as a conditional link predictor, evaluating the compatibility of directed transitions $(v_i, v_j)$ within the context of the task.

*Architecture.* For every candidate edge allowed by the *wGraph* topology, we construct a composite feature vector $\mathbf{z}_{ij}$ by concatenating the source, target, and task embeddings:

$$\mathbf{z}_{ij} = \text{CONCAT}[\mathbf{h}_i, \mathbf{h}_j, \mathbf{h}_{\text{task}}] \in \mathbb{R}^{3D}. \tag{11}$$

This vector is processed by a 3-layer MLP (Linear → ReLU → Linear → ReLU → Linear) to compute the unnormalized compatibility logit $\omega_{ij}$. The probability score $s_{i,j}$ used in Eq. (6) is derived via a sigmoid activation: $s_{i,j} = \sigma(\omega_{ij})$.

*Training: Differentiable Relaxation.* To enable end-to-end gradient-based optimization over discrete structural decisions, we employ the Gumbel-Sigmoid reparameterization trick (Jang et al., 2016) during training. Edge selection is approximated as:

$$\tilde{s}_{i,j} = \sigma\left(\frac{\omega_{ij} + g_{ij}}{\tau}\right), \quad g_{ij} \sim \text{Gumbel}(0,1), \tag{12}$$

where $\tau$ is a temperature hyperparameter. This relaxation bridges the gap between the discrete graph topology and continuous gradient updates (Fu et al., 2026; Wang et al., 2026).

*Inference: Constrained Decoding.* During inference, we bypass the stochastic relaxation and use the deterministic scores $s_{i,j}$ directly. As described in Section 4.2, these scores guide the progressive workflow construction algorithm, which greedily selects high-confidence transitions while strictly enforcing the structural validity constraints of the underlying *wGraph*, rather than sampling an unconstrained adjacency matrix.

### B.3. Training Loss and Optimization of Workflow Construction Model

We adopt an offline supervision framework to train the workflow construction model. Specifically, we collect a dataset $\mathcal{D} = \{(Z_k, \mathcal{W}_k^*)\}_{k=1}^K$, where $Z_k$ denotes a task instance and $\mathcal{W}_k^*$ is a high-quality workflow mined from offline runs, exhibiting strong empirical performance under task-specific evaluation.

The training objective is to encourage the model to produce workflows that are structurally consistent with the target workflows. For each task $Z_k$, we convert the target workflow $\mathcal{W}_k^*$ into binary supervision signals over candidate operation dependencies. Specifically, for each candidate edge $(v_i, v_j)$ permitted by *wGraph*, we define the label

$$\hat{s}_{i,j}^k = \begin{cases} 1, & \text{if } (v_i, v_j) \in \mathcal{E}(\mathcal{W}_k^*), \\ 0, & \text{otherwise}, \end{cases} \tag{13}$$

where $\mathcal{E}(\mathcal{W}_k^*)$ denotes the set of operation dependencies included in the target workflow.

Given the task-conditioned graph constructed with task $Z_k$, the workflow construction model predicts a task-aware compatibility score $s_{i,j}^k \in (0, 1)$ for each candidate edge $(v_i, v_j)$. The model is trained to align these predicted scores with the binary edge labels induced by $\mathcal{W}_k^*$. Since workflow construction reduces to a set of independent edge selection decisions conditioned on the task, we adopt a binary cross-entropy loss, which corresponds to maximum likelihood estimation under a Bernoulli model for each candidate operation pair. Concretely, the training objective is defined as:

$$\mathcal{L}(\Theta) = -\frac{1}{K} \sum_{k=1}^K \frac{1}{|\mathcal{E}_{\text{op}}|} \sum_{(i,j) \in \mathcal{E}_{\text{op}}} \left[\hat{s}_{i,j}^k \log s_{i,j}^k + (1 - \hat{s}_{i,j}^k) \log(1 - s_{i,j}^k)\right]. \tag{14}$$

where $\Theta = \{\Theta_{\text{GNN}}, \Theta_{\text{MLP}}\}$ includes all trainable parameters in the GNN and MLP parts. The training procedure is shown in Algorithm 1.

---

**Algorithm 1** Training Procedure of `GraphFlow`

---

**Input** : Global Operation Graph $\mathcal{G}_{\text{op}} = (\mathcal{V}_{\text{op}}, \mathcal{E}_{\text{op}})$, Supervision Dataset $\mathcal{D} = \{(Z_k, \mathcal{W}_k^*)\}_{k=1}^K$, Batch Size $B$, Initial Parameters $\Theta = \{\Theta_{\text{GNN}}, \Theta_{\text{MLP}}\}$, Learning rate $\eta$

**Output** : Optimized parameters $\Theta^*$

**for** *each batch* $\{(Z_k, \mathcal{W}_k^*)\}_{k=1}^B \subset \mathcal{D}$ **do**

  /* Phase 1: Task-Conditioned Graph Construction (Sec. 4.1) */

  Initialize operation features and query embedding via $\phi(\cdot)$:

  **for** *node $i$ in* $\{1, 2, \cdots, |\mathcal{V}_{\text{op}}|\}$ **do**

    |  $\mathbf{x}_i \leftarrow \phi(v_i)$

  **end**

  Initialize task node $v_{\text{task}}$ with query features derived from $Z_k$:

$$\mathbf{x}_{\text{task}} \leftarrow \phi(Z_k)$$

  Construct task-conditioned graph $\mathcal{G}$ by augmenting $\mathcal{G}_{\text{op}}$ with $v_{\text{task}}$ and connecting edges.

  Form feature matrix $\mathbf{X}$ and adjacency matrix $\mathbf{A}$.

  /* Phase 2: Edge Probability Prediction (Sec. 4.2) */

  Encode structure-aware representations via GNN:

$$\mathbf{H} \leftarrow \text{GNN}(\mathbf{X}, \mathbf{A}; \Theta_{\text{GNN}})$$

  Compute task-aware compatibility scores $s_{i,j}$ for all candidate edges $(v_i, v_j) \in \mathcal{E}_{\text{op}}$ (training-time edge scoring):

$$s_{i,j} \leftarrow \text{Sigmoid}\left(\text{MLP}([\mathbf{h}_i, \mathbf{h}_j, \mathbf{h}_{\text{task}}]; \Theta_{\text{MLP}})\right)$$

  /* Phase 3: Supervised Optimization (Sec. B.3) */

  Construct binary supervision labels $\hat{s}_{i,j}$ from target workflow $\mathcal{W}_k^*$:

$$\hat{s}_{i,j} \leftarrow \begin{cases} 1, & \text{if } (v_i, v_j) \in \mathcal{E}(\mathcal{W}_k^*) \\ 0, & \text{otherwise} \end{cases}$$

  Compute Binary Cross-Entropy loss $\mathcal{L}$:

$$\mathcal{L} \leftarrow -\frac{1}{B}\sum_{k=1}^B \frac{1}{|\mathcal{E}_{\text{op}}|} \sum_{(i,j)\in\mathcal{E}_{\text{op}}} \left[ \hat{s}_{i,j}^{(k)} \log s_{i,j}^{(k)} + (1 - \hat{s}_{i,j}^{(k)}) \log(1 - s_{i,j}^{(k)}) \right]$$

  Update parameters via gradient descent:

$$\Theta \leftarrow \Theta - \eta \nabla_\Theta \mathcal{L}$$

**end**

---

## C. Additional Experimental Results and Analysis

### C.1. Sensitivity Analysis of GNN Encoder Depth

**Optimal Depth for Structural Reasoning.** The results consistently indicate that a 2-layer GCN yields the best performance across all tasks and backbones. Increasing the encoder depth from 1 to 2 layers leads to a clear performance improvement on all benchmarks. For example, on the MATH benchmark with Qwen-2.5-7B, accuracy improves from approximately 70% to 74.5%. This suggests that while a single message-passing layer primarily captures local dependencies, two layers provide a sufficient receptive field to propagate task semantics from the virtual task node throughout the *wGraph*.

**Performance Plateau and Over-smoothing.** Further increasing the GCN depth to 3 layers results in a performance plateau,

whereas a 4-layer configuration leads to a noticeable degradation across benchmarks. For instance, on the Gemma-2-9B backbone, the pass@1 score on MBPP drops from 63.8% to below 59% when increasing the depth from 2 to 4 layers. This trend is consistent with the over-smoothing phenomenon commonly observed in deep GNNs, where node representations become increasingly homogenized with additional message-passing steps, reducing the discriminative capacity required for accurate workflow construction.

**Generalization Across Backbones.** The consistent optimal depth across different LLM backbones suggests that the structural encoding capacity of `GraphFlow` is robust to variations in model scale and architecture. Overall, a 2-layer GCN provides an effective inductive bias for mapping diverse task queries to task-specific subgraphs within the proposed framework.

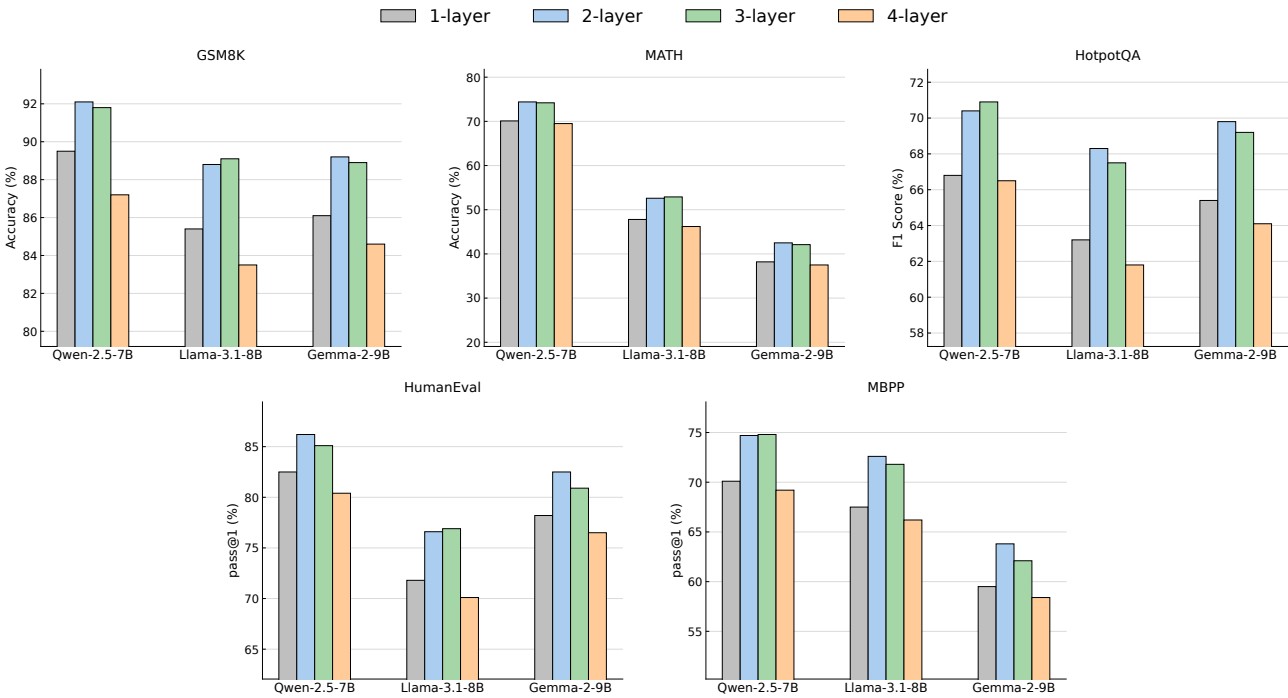

*Figure 8.* Ablation study on GCN depth. We evaluate the performance impact of varying GCN layers (from 1 to 4) across five benchmarks. The results demonstrate that the 2-layer architecture consistently yields optimal performance across different LLM backbones.

### C.2. Ablation Study on Workflow Generation

To verify the effectiveness of the core components in `GraphFlow`, we conduct a comprehensive ablation study using the Llama-3.1-8B backbone across all five benchmarks. The results, summarized in Table 2, evaluate two distinct variants: (1) *w/o* GNN, where the graph encoder is replaced by a semantic-only MLP that processes operations in isolation, discarding topological message passing; and (2) *w/o* $v_{\text{task}}$, which removes the virtual task node, relying solely on operation features without global task-conditioned message diffusion.

The ablation results highlight the importance of topology-aware modeling in workflow generation. Replacing the GNN-based structural encoder with a semantic-only MLP leads to the largest performance degradation across all benchmarks, including a 5.5% accuracy drop on MATH (52.6% → 47.1%) and a 5.3% drop on GSM8K. These results indicate that explicitly modeling structural dependencies among operations is critical for complex, multi-step reasoning. While introducing the GNN encoder incurs a modest increase in P90 inference latency (e.g., from 15.20s to 15.95s on MATH), the resulting gains in execution quality substantially outweigh this overhead. Furthermore, removing the virtual task node $v_{\text{task}}$ leads to consistent but smaller performance declines (e.g., 2.3% on MATH), demonstrating that globally diffusing task-specific information across the operation graph is essential for synthesizing contextually relevant workflow subgraphs.

*Table 2.* Component ablation results for task-adaptive workflow generation using the Llama-3.1-8B backbone.

| Method Variant | GSM8K | | MATH | | HotpotQA | | HumanEval | | MBPP | |
|---|---|---|---|---|---|---|---|---|---|---|
| | Acc. | Time | Acc. | Time | F1 | Time | pass@1 | Time | pass@1 | Time |
| **Vanilla GraphFlow** | 88.8 | 8.85 | 52.6 | 15.95 | 68.3 | 2.38 | 76.6 | 13.45 | 72.6 | 15.15 |
| GraphFlow *w/o* GNN | 83.5 | 8.50 | 47.1 | 15.20 | 63.5 | 2.15 | 69.8 | 12.90 | 65.4 | 14.60 |
| GraphFlow *w/o* $v_{\text{task}}$ | 86.2 | 8.70 | 50.3 | 15.83 | 66.1 | 2.35 | 73.5 | 13.25 | 69.8 | 15.00 |

## D. Case Studies: Atomic Operations and Workflows

This section provides concrete examples of the structural components within the GraphFlow framework. We first present representative atomic operations characterized by their semantic instructions and trigger patterns, followed by a graph-structured workflow demonstrating the dependency logic between these operations.

### D.1. Examples of Atomic Operations

---
**Operation: Equation Translation**

**ID:** OP_MATH_01
**Instruction ($\mathcal{D}_i$):**
Translate the natural language story or relationship statement into a formal linear equation.
**Trigger Patterns ($\mathcal{P}_i$):**
- MUST: ["solve", "unknown", "find", "total"]
- SHOULD: ["let x", "value of unknown x"]

---
**Operation: Variable Isolation**

**ID:** OP_MATH_02
**Instruction ($\mathcal{D}_i$):**
Isolate the variable by applying inverse operations to both sides of the linear equation.
**Trigger Patterns ($\mathcal{P}_i$):**
- MUST: ["calculate", "=", "left", "remain"]
- SHOULD: ["x", "result"]

---

### D.2. Example of a Workflow

**Example of a Workflow for Multi-variable Algebraic Reasoning**

```json
{
  "id": "WF_MATH_001",
  "name": "Systemic Algebraic Solver",
  "description": "Handles word problems requiring variable assignment, system
      formulation, and iterative verification.",
  "patterns": {
    "must": ["system", "equations", "variables", "solve"],
    "should": ["simultaneous", "substitution", "elimination"]
  },
  "graph_structure": {
    "nodes": ["OP_01", "OP_02", "OP_03", "OP_04", "OP_05"],
    "edges": [
      ["OP_01", "OP_02"],
```

```
        ["OP_01", "OP_03"],
        ["OP_02", "OP_04"],
        ["OP_03", "OP_04"],
        ["OP_04", "OP_05"]
    ]
  },
  "operations": {
    "OP_01": {
      "name": "Variable Mapping",
      "instruction": "Identify all unknown quantities and assign unique symbolic
          variables (e.g., x, y, z).",
      "patterns": {"must": ["unknown", "assign"], "should": ["let"]}
    },
    "OP_02": {
      "name": "Relationship Extraction",
      "instruction": "Extract primary constraints from the text and formulate them
          into linear equations.",
      "patterns": {"must": ["equation", "formulate"], "should": ["sum", "total"]}
    },
    "OP_03": {
      "name": "Constraint Cross-Check",
      "instruction": "Identify implicit constraints (e.g., non-negativity) not
          explicitly stated in the problem.",
      "patterns": {"must": ["constraint", "implicit"], "should": ["range", "limit"]}
    },
    "OP_04": {
      "name": "System Solver",
      "instruction": "Solve the system of equations using substitution or elimination
          techniques.",
      "patterns": {"must": ["solve", "calculate"], "should": ["value of x"]}
    },
    "OP_05": {
      "name": "Logical Reality Check",
      "instruction": "Verify if the numerical solution makes sense in the real-world
          context of the problem.",
      "patterns": {"must": ["verify", "check"], "should": ["correct", "logical"]}
    }
  }
}
```

# E. Extended Analysis of KV Cache Sparsity

### E.1. Experimental Details for KV Sparsity Analysis

To empirically validate the sparsity of prefix-induced KV cache changes, we conduct a series of controlled experiments. The key experimental configurations are summarized as follows:

- **Model and Hardware:** We utilize the Llama-3.1-8B model (32 layers, 32 heads) in `BFloat16` precision. All measurements are performed on an NVIDIA L20 GPU.
- **Token Statistics:** The analysis uses a contextualized sequence of approximately 200 tokens, consisting of a document-based prefix (135 tokens) and a target operation instruction (65 tokens). We compare the KV states of the target operation computed with and without the prefix, using position ID alignment to ensure consistency.
- **Sampling Strategy:** We extract KV residuals from a representative subset of layers ($\{5, 10, 15, 20, 25, 30\}$) and attention heads ($\{0, 3, 6\}$) to capture cross-network behavior.
- **Sparsity Metric:** We measure effective sparsity, defined as the percentage of elements that can be pruned while retaining 95% of the total energy (Frobenius norm) of the residual tensor $\Delta \mathbf{KV}$.

### E.2. Additional Visualizations of Residual KV Patterns

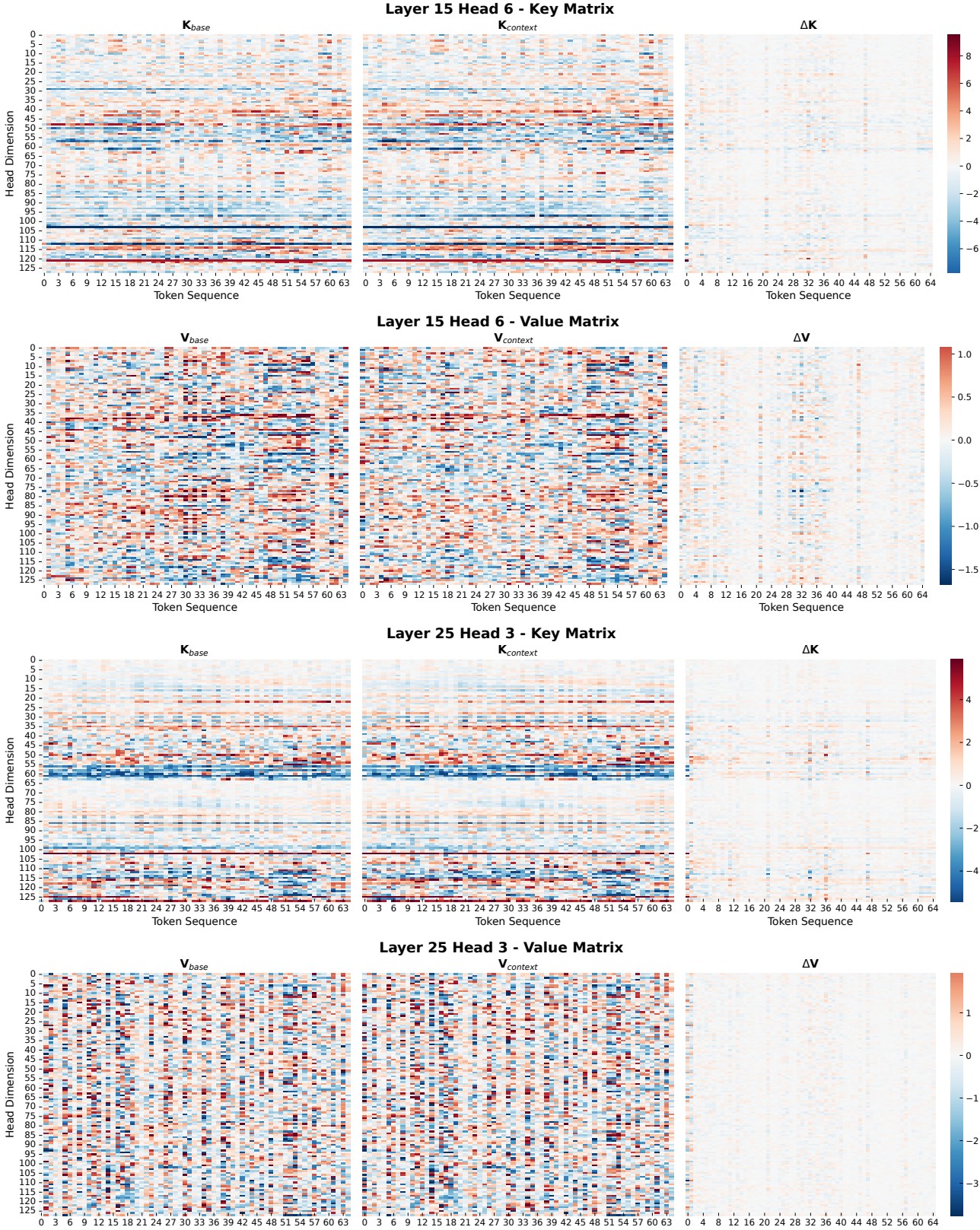

*Figure 9.* Additional visualizations of residual KV sparsity. We show the element-wise difference heatmaps for keys ($\Delta K$) and values ($\Delta V$) at representative layers and attention heads, comparing contextualized KV states against the context-free base cache.

