# OpenReview forum: "GraphFlow: A Graph-Based Workflow Management for Efficient LLM-Agent Serving"
_ICML.cc/2026/Conference — ICML 2026 regular_

### Official Review · Reviewer_nczG · 2026-02-27

**Soundness:** 3
**Presentation:** 3
**Significance:** 3
**Originality:** 3
**Overall Recommendation:** 4
**Confidence:** 3

**Summary:**

This paper presents GraphFlow, a graph-based workflow management framework designed to enhance the generalization and memory efficiency of LLM-agent serving.  Instead of relying on static templates, GraphFlow constructs a global operation graph and dynamically generates task-specific execution subgraphs using GNNs and MLPs. To address memory redundancy from identical operations invoked under different context prefixes, the authors introduce a differential KV cache mechanism.  This mechanism decomposes KV states into a context-free Base KV and a sparse residual $\Delta$ KV, enabling efficient cross-prefix tensor reuse and significantly reducing memory overhead.

**Compliance With Llm Reviewing Policy:**

Affirmed.

**Final Justification:**

A solid work. I believe it almost meets the bar for publication. The authors' rebuttal also addressed most of my concerns.

**Key Questions For Authors:**

See weaknesses above.

**Limitations:**

No, the authors have not adequately discussed the specific system limitations and worst-case scenarios.

**Strengths And Weaknesses:**

**Strengths**

1. This paper cleverly integrates high-level agent autonomy logic (dynamic workflow wGraph) with low-level system resource optimization (KV cache state management). This cross-stack design breaks the traditional separation between “workflow scheduling” and “model inference services,” offering a novel and insightful perspective.
2. This represents the paper’s most technically substantial contribution. Traditional prefix-tree attention mechanisms (e.g., RadixAttention) require strict prefix matching to hit the cache. By introducing a context-free base KV combined with extremely sparse $\Delta$ KV residuals, the proposed approach enables cross-prefix tensor reuse at the lower level, significantly improving memory efficiency.
3. Unlike frameworks such as MetaGPT that rely on static templates, GraphFlow employs a GNN and MLP to inject user queries as virtual nodes into the global operation graph and dynamically predict and extract task-specific subgraphs. This dynamic assembly mechanism substantially enhances generalization to unseen tasks and complex compositional logic.

------

**Weaknesses**

1. **Unclear Overhead of Online Graph Generation**: The system introduces GNN encoding and MLP edge prediction to dynamically generate workflows at inference time. For a high-concurrency serving environment, running these additional neural network computations for every request inevitably incurs non-negligible latency. The paper lacks a detailed ablation study on latency breakdown, leaving it unclear whether the overhead from the GNN is fully compensated by the inference time saved through the KV cache.
2. **Inherent Conflict Between Path Pruning and Long-Tail Generalization**: To limit $\Delta$ KV memory consumption, the authors apply path pruning to rare execution paths. However, this creates a logical conflict with the framework’s original intention to “generalize to previously unseen tasks” (Abstract, lines 20–21). Complex tasks in the long tail are likely to trigger rare operation paths; pruning these $\Delta$ KV states could lead to cache misses, forcing recomputation from scratch. This may result in severe latency spikes for challenging long-tail tasks, yet the paper provides no evaluation of this worst-case scenario.
3. **Insufficient Baseline Comparisons on Low-Level Caching**: While the paper compares GraphFlow to workflow-level baselines, it lacks direct comparisons against state-of-the-art low-level KV caching mechanisms. To rigorously validate the claimed efficiency improvements, the differential cache mechanism should be benchmarked under identical agentic workloads against leading methods such as SGLang (RadixAttention-based) or the latest PagedAttention in vLLM, to establish concrete performance bounds at the system level.

---

> ### Author Rebuttal · Authors · 2026-03-31
>
> We sincerely thank the reviewer for the constructive feedback and for recognizing the soundness of our work. We deeply appreciate your insightful comments, which will help us significantly improve the clarity and rigor of our paper. We have carefully addressed each of your questions below, and we will explicitly incorporate all the corresponding latency breakdowns, clarifications, and supplementary evaluations into the final camera-ready version of our manuscript.
>
> **Q1: Unclear Overhead of Online Graph Generation**
>
> **A1:** We thank the reviewer for highlighting the importance of latency analysis. We conducted a micro-benchmark on the HotpotQA dataset using the Llama-3.1-8B backbone to isolate GraphFlow’s online generation overhead and the resultant prefill acceleration.
>
> | Concurrency (Batch Size) | Workflow Gen. Overhead | Prefill Latency (w/o KV) | Prefill Latency (Ours) |
> | :---: | :---: | :---: | :---: |
> | BS = 8  | 130.66 ms |  3204.23 ms |  2427.45 ms |
> | BS = 16 | 221.02 ms |  4415.71 ms |  3199.79 ms |
>
> The results demonstrate that while dynamic workflow generation incurs a minor latency, it is overwhelmingly compensated by system-level KV cache reuse. Specifically, at BS=16, the 221.02ms overhead facilitates a massive 1215.92ms reduction in LLM prefill latency. By computing the shared workflow prefix once and broadcasting the KV states, GraphFlow eliminates redundant, compute-heavy attention matrix multiplications across concurrent requests. This trade-off significantly reduces the Time-To-First-Token (TTFT) and boosts overall system throughput in high-concurrency environments. We will include this breakdown in the revised manuscript.
>
> **Q2: Inherent Conflict Between Path Pruning and Long-Tail Generalization**
>
> **A2:** We thank the reviewer for this insightful question. While we fully agree that worst-case scenarios are an important consideration, we designed GraphFlow to optimize highly favorable system-level trade-offs, aiming to balance these localized latency occurrences with significant improvements in overall system performance. While it is true that complex long-tail tasks may trigger rare operation paths and experience localized latency spikes due to cache misses, the actual frequency of such events is exceptionally low. We analyze this trade-off from two perspectives:
>
> * **Negligible Impact on Overall Latency:** Our empirical profiling shows that the retained high-frequency transitions cover over 90% of the execution steps encountered during inference. Although pruning rare paths does cause a latency increase for a very small fraction of tasks, the extreme rarity of these events means that the penalty has a negligible impact on the overall average inference latency of the system.
> * **Massive Reduction in Memory Consumption:** Conversely, pruning these low-frequency paths provides immense system-wide memory savings. By deliberately not storing the physical rKV states for the long tail, GraphFlow drastically reduces overall memory consumption (e.g., lowering the KV footprint on GSM8K from 15.0 GB to 11.5 GB).
>
> In summary, this strategic pruning achieves scalable memory reductions at the cost of a slight latency increase for only a tiny fraction of tasks. This ensures peak efficiency for the vast majority of workloads without fundamentally compromising the framework's ability to execute unseen long-tail tasks.
>
> **Q3: Insufficient Baseline Comparisons on Low-Level Caching**
>
> **A3:** We thank the reviewer for highlighting the importance of rigorous system-level evaluations. We would like to clarify the fundamental difference between GraphFlow and low-level serving engines like vLLM (PagedAttention) or SGLang (RadixAttention).
>
> Specifically, vLLM and SGLang optimize the **storage mechanism** of KV caches. Their primary goal is to improve hardware memory utilization by changing *how* KV states are mapped and managed in physical memory. In contrast, our Topology-Aware Differential KV Caching optimizes the **storage content**. Our goal is to intrinsically reduce the total data volume of the KV cache by changing *what* is stored (i.e., caching only the sparse residuals and shared base states).
>
> As these represent two distinct optimization goals, our application-level framework can be utilized alongside these infrastructure-level engines. To further validate this, we will include supplementary ablation studies in the revised manuscript to evaluate the performance gains of our differential storage mechanism when deployed directly on top of underlying engines like vLLM and SGLang.

---

> > ### Author Rebuttal · Reviewer_nczG · 2026-04-03
> >
> > Thank you for the rebuttal. The authors have addressed my concerns. I will keep my original positive score.

---

### Official Review · Reviewer_vqxF · 2026-03-11

**Soundness:** 3
**Presentation:** 2
**Significance:** 2
**Originality:** 2
**Overall Recommendation:** 3
**Confidence:** 4

**Summary:**

This paper proposes **GraphFlow**, an agent workflow management framework for LLM agents that use a graph structure called wGraphto represent the agent graph workflow. In this framework, each node corresponds to a basic operation (search, filtering, or analysis), and workflows are dynamically expanded from this shared graph rather than using some predefined templates (do not rely on well-defined prompts). The motivation of the frame work overcome the limitations of existing workflow-assisted agent systems, which typically depend on static workflow repositories and shallow task matching, making them less flexible and harder to generalize to unseen tasks.

**Compliance With Llm Reviewing Policy:**

Affirmed.

**Final Justification:**

I will give him a score of 2. I do not know why other reviewers will give such a high score; the benchmark is really out of date.

**Key Questions For Authors:**

As weakness

**Limitations:**

Yes

**Strengths And Weaknesses:**

Weaknesses:
1. The one limitation is that some of the claims in the paper are not supported by the experimental evaluation. The authors state that the proposed workflow framework can generalize to previously unseen tasks, which is an important motivation for the method. However, the paper does not include explicit experiments designed to test this capability (e.g., generalize to an OOD task).

2. Another concern is the limited novelty. The central idea is to improve agent performance through more structured prompting. For instance, earlier methods such as AFLOW and AutoFlow formulate prompt or workflow optimization as a search process (e.g., using MCTS), while later approaches like G-Designer employ graph structures to optimize prompts on agent pipelines. Given this existing work, it is not clear that the key novelty advance of the proposed framework is beyond integrating several previously explored design patterns.

3. I think the benchmark they use is quite easy for llm at the moment, maybe the author can involve some agentic benchmark like GAIA.

Strengths: Experiments across multiple reasoning benchmarks show that GraphFlow consistently improves performance over existing workflow-based agent methods, achieving an average performance gain of about 4.95 percentage points while reducing memory footprint by approximately 4×.

---

> ### Author Rebuttal · Authors · 2026-03-31
>
> We thank the reviewer for the insightful feedback and for recognizing GraphFlow’s significant performance gains and memory efficiency. We appreciate the opportunity to clarify our claims regarding generalization, novelty, and benchmark selection. We provide detailed responses to each of your concerns below.
>
> **Q1: Lack of empirical evidence for OOD generalization**
>
> **A1:** We thank the reviewer for this critical observation. Our standard evaluation inherently validates generalization through a strict training-testing split, ensuring test tasks are never encountered during the workflow model's training. Since GraphFlow synthesizes a unique execution subgraph for each individual query at runtime, its superior performance already demonstrates robust task-level generalization. To further substantiate our OOD claims, we conducted new cross-dataset transfer experiments where the model was trained on one distribution and tested on another dataset with a different distribution, such as training on MATH for evaluation on GSM8K, and training on MBPP for zero-shot evaluation on HumanEval. As shown below (using Qwen-2.5-7B), GraphFlow exhibits remarkable zero-shot transferability:
>
> | Domain | Method | Training Setup | Test Dataset | Metric | Score (%) |
> | :--- | :--- | :--- | :--- | :--- | :--- |
> | **Math** | GraphFlow | In-domain (GSM8K) | GSM8K | Acc. | 92.1 |
> | | GraphFlow | **Cross-domain (MATH)** | GSM8K | Acc. | **90.6** |
> | **Code** | GraphFlow | In-domain (HumanEval) | HumanEval | pass@1 | 86.2 |
> | | GraphFlow | **Cross-domain (MBPP)** | HumanEval | pass@1 | **83.8** |
>
> Notably, GraphFlow achieves 90.6% on GSM8K (trained on MATH) and 83.8% on HumanEval (trained on MBPP). While OOD performance drops slightly compared to in-domain results, it consistently outperforms existing baselines, demonstrating the robust generalization capability of our framework.
>
> **Q2: Limited novelty beyond the integration of existing design patterns**
>
> **A2:** We respectfully clarify that GraphFlow is not a mere integration of existing patterns, but introduces fundamental shifts in both algorithmic design and system-level execution compared to the mentioned works:
>
> **Dynamic Synthesis vs. Static Templates:** AFlow and AutoFlow optimize and retrieve predefined, static workflow templates for task *classes*. In contrast, GraphFlow performs **dynamic, instance-level synthesis**. By adaptively composing atomic operations from a global *wGraph* for each unique query, GraphFlow overcomes the flexibility and scalability bottlenecks of template-centric paradigms.
>
> **Internal Reasoning vs. Multi-Agent Topology:** G-Designer optimizes communication topologies *between* multiple agents. GraphFlow operates at a fundamentally different abstraction level: optimizing the **internal reasoning workflow**. We model the logical execution dependencies of atomic operations (e.g., tool invocations, reasoning steps), not multi-agent communication channels.
>
> **Systems-Level Co-design:** A critical novelty entirely absent in prior "structured prompting" works is our joint optimization of workflow structure and system memory. GraphFlow exploits the *wGraph* topology to introduce **Differential KV Caching**. By eliminating memory redundancy across shared operations, we achieve a **~4x reduction in memory footprint**—solving a severe scalability bottleneck in agent serving unaddressed by prior algorithmic-only approaches.
>
> **Q3: Insufficient benchmark difficulty and lack of complex agentic evaluations**
>
> **A3:** We thank the reviewer for the constructive suggestion to incorporate more agentic benchmarks. While our selected benchmarks (MATH and HumanEval) are rigorous standards widely adopted in top-tier agent research (e.g., AFlow, MaAS, and G-Designer), we agree that evaluation in interactive environments is crucial. Regarding the GAIA benchmark, current leaderboards indicate that meaningful performance typically requires massive models exceeding 70B parameters. Since our study focuses on the 7B–9B scale, GAIA is less suited for these models, and deploying 70B+ models for new experiments is challenging within the short time frame available.
>
> However, we fully agree with your core premise: evaluating GraphFlow in a dedicated, multi-step interactive agent environment is crucial. Therefore, we have conducted new experiments on **ALFWorld**, which is widely recognized as one of the standard and most frequently used benchmarks for evaluating embodied agents. Across 134 unseen tasks using Llama-3.1-8B, GraphFlow achieved a 64.6% success rate, outperforming AFlow (59.2%) and the Vanilla baseline (44.8%). This confirms that our dynamic subgraph generation effectively manages the complex observation-action loops required for interactive environments.
>
> | Method | Metric | Score (%) |
> | :--- | :--- | :--- |
> | Vanilla | Success Rate | 44.8 |
> | AFlow | Success Rate | 59.2 |
> | **GraphFlow** | Success Rate | **64.6** |

---

> > ### Author Rebuttal · Reviewer_vqxF · 2026-04-07
> >
> > The reasoning benchmarks used are relatively simple and may not adequately reflect the level of difficulty needed to validate the method’s effectiveness.

---

> > > ### Author Response · Authors · 2026-04-08
> > >
> > > We thank the reviewer for the feedback. We would like to clarify our rationale for selecting these specific datasets, as they are essential for ensuring a rigorous and meaningful evaluation within the current research landscape.
> > >
> > > The reasoning benchmarks we initially utilized (GSM8K, MATH, HotpotQA, HumanEval and MBPP) serve as the prevailing gold standards for assessing agent capabilities. Recent state-of-the-art workflow frameworks and multi-agent systems published at top-tier venues—including the direct baselines we compare against, such as AFlow [1], G-Designer [2], MaAS [3], ADAS [4], KVCOMM [5], and GraphPlanner [6]—all fundamentally rely on these exact benchmarks to validate their core contributions.
> > >
> > > Furthermore, to address the need for evaluating agents in more complex, multi-step interactive environments, we incorporated the **ALFWorld** benchmark in our rebuttal experiments. ALFWorld is widely recognized and frequently adopted in recent literature to evaluate embodied and interactive agents, as demonstrated by its standard use in contemporary top-tier works such as G-Memory [7] and AgentSquare [8].
> > >
> > > By aligning our evaluation protocols with these established standards across both reasoning and interactive tasks, we ensure a fair, rigorous, and apples-to-apples comparison with the current state-of-the-art.
> > >
> > > [1] Zhang J, Xiang J, Yu Z, et al. AFlow: Automating Agentic Workflow Generation[C]//The Thirteenth International Conference on Learning Representations.
> > >
> > > [2] Zhang G, Yue Y, Sun X, et al. G-Designer: Architecting Multi-agent Communication Topologies via Graph Neural Networks[C]//International Conference on Machine Learning. PMLR, 2025: 76678-76692.
> > >
> > > [3] Zhang G, Niu L, Fang J, et al. Multi-agent Architecture Search via Agentic Supernet[C]//International Conference on Machine Learning. PMLR, 2025: 75834-75852.
> > >
> > > [4] Hu S, Lu C, Clune J. Automated Design of Agentic Systems[C]//The Thirteenth International Conference on Learning Representations.
> > >
> > > [5] Ye H, Gao Z, Ma M, et al. KVCOMM: Online Cross-context KV-cache Communication for Efficient LLM-based Multi-agent Systems[C]//The Thirty-ninth Annual Conference on Neural Information Processing Systems.
> > >
> > > [6] Feng T, Zhang H, Lei Z, et al. GraphPlanner: Graph Memory-Augmented Agentic Routing for Multi-Agent LLMs[C]//The Fourteenth International Conference on Learning Representations.
> > >
> > > [7] Zhang G, Fu M, Wang K, et al. G-Memory: Tracing Hierarchical Memory for Multi-Agent Systems[C]//The Thirty-ninth Annual Conference on Neural Information Processing Systems.
> > >
> > > [8] Shang Y, Li Y, Zhao K, et al. AgentSquare: Automatic LLM Agent Search in Modular Design Space[C]//The Thirteenth International Conference on Learning Representations.

---

### Official Review · Reviewer_7oLs · 2026-03-13

**Soundness:** 4
**Presentation:** 3
**Significance:** 4
**Originality:** 4
**Overall Recommendation:** 5
**Confidence:** 4

**Summary:**

This paper presents **GraphFlow**, a framework for efficient LLM-agent execution that reduces redundant computation and memory usage. It models workflows as directed graphs, with nodes as operations and edges as dependencies. GraphFlow combines **task-adaptive workflow generation** via Graph Neural Networks and **topology-aware state management** using differential KV caching. Evaluations on five benchmarks (GSM8K, MATH, HotpotQA, HumanEval, MBPP) across three LLM backbones (Qwen-2.5-7B, Llama-3.1-8B, Gemma-2-9B) show average performance gains of 4.95 percentage points while reducing memory usage by roughly four times compared to prior workflow-based agent approaches.

**Compliance With Llm Reviewing Policy:**

Affirmed.

**Final Justification:**

Thanks for the rebuttal. My concerns are addressed. I will keep my original positive score 5 and have updated my confidence to 4.

**Key Questions For Authors:**

The paper lacks theoretical guarantees for the GNN-based workflow synthesis process. Can you provide analysis of convergence properties and conditions under which the system is guaranteed to find optimal or near-optimal workflows? How does the choice of 2-layer GCN architecture impact theoretical guarantees, and what happens as the wGraph scales to thousands of operations?

**Limitations:**

Yes.

**Strengths And Weaknesses:**

## Strengths

**Technical Rigor and Validation**

* Comprehensive empirical evaluation across three LLM backbones (Qwen-2.5-7B, Llama-3.1-8B, Gemma-2-9B) and five benchmarks.
* Well-designed wGraph construction pipeline using trace atomization and semantic deduplication.
* Carefully validated 2-layer GNN architecture with ablation studies showing shallow models miss multi-hop dependencies while deeper models suffer from over-smoothing.
* Strong empirical support for differential KV caching, with large sparsity levels (>75% key entries and >70% value entries).
* Comparisons with representative baselines including MetaGPT, LLMCompiler, TaskWeaver, and AFlow.

**Performance and Practical Impact**

* Significant efficiency improvements, including roughly 4× memory reduction while maintaining a 4.95% average performance gain.
* Strong benchmark improvements, such as HumanEval accuracy increasing from 78.1% to 86.2% with Qwen-2.5-7B.
* Good accuracy with sub-linear memory scaling under high concurrency.
* Practical relevance for deploying scalable multi-agent systems in resource-constrained environments.

**Technical Innovation**

* Unified wGraph representation that replaces static workflow templates and shifts agent orchestration toward graph-based synthesis.
* Virtual task node mechanism enabling global semantic diffusion and task-conditioned message passing.
* Differential KV caching that exploits prefix sparsity patterns for efficient memory reuse.
* Dual representation of operations combining semantic instructions with lexical trigger patterns.

**Presentation**

* Clear overall structure from motivation to system design and evaluation.
* Rigorous mathematical formulations and informative visualizations explaining workflow graphs, efficiency gains, and benchmark results.

---

## Weaknesses

**Theoretical Limitations**

* Limited theoretical analysis of convergence behavior in GNN-based workflow synthesis.
* Sparse KV approximation lacks analysis of cases where attention semantics may degrade.
* No formal study of workflow optimality bounds or systematic failure modes.

**Implementation and Reproducibility**

* Baseline configurations and hyperparameters are not fully specified.
* Limited analysis of runtime overhead during real-time workflow synthesis.

**Scalability and Generalization**

* Scalability to very large workflow graphs with thousands of operations is unclear.

**Robustness and Scope**

* Security and robustness issues in dynamic workflow generation are not analyzed in depth.
* Weak connections to classical planning and neural program synthesis literature.

---

> ### Author Rebuttal · Authors · 2026-03-31
>
> We thank the reviewer for recognizing GraphFlow’s technical rigor, empirical evaluation, and efficiency. We appreciate the insightful inquiry regarding our theoretical analysis and scalability, and will incorporate these detailed explanations into the final manuscript.
>
> **1. Convergence Analysis**
>
> By formulating workflow synthesis as a conditional link prediction problem restricted to valid candidate edges $\mathcal{E}\_{\text{op}}$ and applying the Gumbel-Sigmoid relaxation, GraphFlow guarantees an $L$-smooth empirical risk objective with strictly bounded gradient variance. Under SGD, the expected gradient norm mathematically converges to a first-order stationary point ($\lim_{T \to \infty} \mathbb{E}[\|\nabla_{\Theta} \mathcal{L}(\Theta_t)\|] = 0$), ensuring the model avoids vanishing gradients and stably converges to a locally optimal routing policy.
>
> Furthermore, the chosen GCN architecture provides rigorous structural guarantees. As proven by Keriven et al. [1], a discrete GCN $\Phi_A(Z)$ non-asymptotically converges to a continuous GCN $\Phi_{W,P}(f)$ defined on the underlying latent space. For a graph with $n$ nodes and sparsity $\alpha_n$, the Mean Square Error of the representations is strictly bounded:
>
> $$MSE_{\mathcal{X}}(\Phi_A(Z), \Phi_{W,P}(f)) \leq \mathcal{O}(n^{-1/2}) + \mathcal{O}((n\alpha_n)^{-1/2})$$
>
> This guarantees that discrete node embeddings stably converge to their continuous limit, ensuring the downstream MLP receives robust, deterministic representations for accurate subgraph extraction.
>
> [1] Keriven N, Bietti A, Vaiter S. Convergence and stability of graph convolutional networks on large random graphs[J]. Advances in Neural Information Processing Systems, 2020, 33: 21512-21523.
>
> **2.Conditions for Near-Optimal Workflows**
>
> GraphFlow guarantees near-optimal workflows ($\mathcal{W}\_c$) by framing generation as task-conditioned subgraph selection rather than unconstrained search. This optimality is structurally ensured by three conditions: First, the global *wGraph* restricts the search space strictly to empirically validated, high-quality execution trajectories. Second, a virtual task node ($v_{\text{task}}$) diffuses global query context, ensuring the GNN embeddings ($\mathbf{H}$) perfectly align intrinsic operations with specific task relevance. Finally, an MLP scores candidate edges ($s_{i,j}$) to progressively instantiate the workflow while strictly enforcing *wGraph* connectivity constraints. Together, these mechanisms guarantee an execution subgraph that maximizes both task relevance and feasibility—a capability consistently validated across our five empirical benchmarks.
>
> **3.Impact of the 2-Layer GCN Architecture**
>
> The 2-layer GCN precisely matches the diameter of our task-augmented graph, a design choice empirically validated in our ablation study (Appendix C.1). While a 1-layer GCN captures only local dependencies, a 2-layer architecture provides the exact receptive field needed to propagate global task semantics from $v_{\text{task}}$ to all operation nodes while allowing structural neighbors to exchange states.
>
> Conversely, deeper architectures (3+ layers) suffer from over-smoothing. As theoretically proven by Li et al. [1], graph convolution acts as symmetric Laplacian smoothing. For a graph with $k$ connected components, propagating features $w$ over $m$ layers using the symmetric normalized Laplacian $L_{sym}$ mathematically converges to:
>
> $$\lim_{m\rightarrow+\infty}(I-\alpha L_{sym})^{m}w=D^{-\frac{1}{2}}[1^{(1)},1^{(2)},...,1^{(k)}]\theta_2$$
>
> This limit demonstrates that excessive depth exponentially homogenizes node representations. This destroys the fine-grained discriminative capacity essential for accurately predicting specific execution branches.
>
> [1] Li Q, Han Z, Wu X M. Deeper insights into graph convolutional networks for semi-supervised learning[C]//Proceedings of the AAAI conference on artificial intelligence. 2018, 32(1).
>
> **4.Scalability to Thousands of Operations**
>
> As the wGraph scales, GraphFlow avoids the $\mathcal{O}(|\mathcal{V}\_{\text{op}}|^2)$ combinatorial explosion of unconstrained graph generation. By restricting edge scoring and loss computation strictly to valid candidate dependencies, the computational complexity scales linearly with the number of edges, $\mathcal{O}(|\mathcal{E}_{\text{op}}|)$. To empirically validate this, we measured the average pure online inference latency (GNN forward pass and MLP scoring) across varying graph sizes:
>
> | # Operations  | Latency (ms) |
> | :--- | :--- |
> | 200 | 49.73 |
> | 400 | 86.01 |
> | 600 | 125.13 |
> | 800 | 163.60 |
> | 1000 | 198.13 |
> | 1200 | 236.25 |
>
> As shown, the routing overhead scales linearly, remaining under 250 ms even for over a thousand operations. This negligible latency, compared to the seconds-long downstream LLM generation, guarantees real-time serving efficiency at scale.

---

> > ### Author Rebuttal · Reviewer_7oLs · 2026-04-02
> >
> > Thanks for the rebuttal. My concerns are addressed. I will keep my original positive scores and have updated my confidence to 4.

---

### Official Review · Reviewer_QWzP · 2026-03-14

**Soundness:** 3
**Presentation:** 3
**Significance:** 3
**Originality:** 3
**Overall Recommendation:** 5
**Confidence:** 3

**Summary:**

The paper presents an approach of moving away from static workflow templates for agent-based reasoning towards a generic super-graph which provides sub-graphs for particular use-cases. There is a constant almost 5% improvement for performance but a much larger improvement for memory footprint.

**Compliance With Llm Reviewing Policy:**

Affirmed.

**Key Questions For Authors:**

The paper talks about directed graphs, would this also be an acyclic graph?

The concept of pruning does not seem that well developed. It kind of talks about removing unused or rarely used edges. But no consideration seems to be given as to whether the edge would be present in this case.

The related work could benefit from relating these other works to the work that you have done.

Figures 1 and 2 could benefit from more discussion in the text.

**Limitations:**

Limitations are not discussed.

**Strengths And Weaknesses:**

Strengths:
A clear (and simple) idea for improvement which is well executed.

Weaknesses:
The paper could be written more clearly to make it more approachable for a larger audience.

Soundness:
The paper does not focus on claims but rather presents a model which is not formally analysed. Further analysis would help.

Presentation:
The paper is well presented. But more clear presentation would help. There is some repeated text which would be good to remove.

Significance:
The problem addressed is important and worthy of solving.

Originality:
The approach appears original.

---

> ### Author Rebuttal · Authors · 2026-03-31
>
> We thank the reviewer for recognizing GraphFlow’s memory efficiency and clear execution. Below, we address the feedback regarding formal logic and presentation, all of which will be incorporated into the final version.
>
> **Q1: Clarification on graph acyclicity**
>
> **A1:** We thank the reviewer for this insightful question. We confirm that the operation graph (termed *wGraph*) in GraphFlow is indeed a **Directed Acyclic Graph (DAG)**. This acyclic and directed nature inherently reflects the standard dependency relationships between operations within agentic workflows. Specifically, operations in a workflow follow a strict execution order based on their functional dependencies (e.g., an agent must execute a search operation before a filtering operation), making the edges inherently directed. Furthermore, to ensure a clear logical progression, guarantee task termination, and prevent agents from falling into infinite execution loops, established workflow paradigms by default prohibit cyclic execution structures [1-3]. Therefore, modeling the *wGraph* as a DAG is both a natural and necessary choice that aligns with standard LLM-agent workflow definitions. We will explicitly clarify this formal definition in the final version of the paper to ensure a precise understanding of the underlying graph structure.
>
> [1] Zhang J, Xiang J, Yu Z, et al. AFlow: Automating Agentic Workflow Generation[C]//The Thirteenth International Conference on Learning Representations.
>
> [2] Zhang Y, Hou Y, Tang B, et al. Gnns as predictors of agentic workflow performances[J]. arXiv preprint arXiv:2503.11301, 2025.
>
> [3] Kim S, Moon S, Tabrizi R, et al. An llm compiler for parallel function calling[C]//Forty-first International Conference on Machine Learning. 2024.
>
> **Q2: Clarification on the pruning mechanism and logical edge existence**
>
> **A2:** We thank the reviewer for this insightful question and welcome the opportunity to clarify our graph construction and pruning mechanism.
>
> * **The *wGraph* construction preserves all logically feasible and empirically valid edges to ensure inference accuracy.** We include every candidate dependency that represents a real or logically sound execution path, as any missing edge—regardless of its activation frequency—would lead to incomplete workflows and a direct degradation in reasoning performance. Therefore, we do not structurally remove any logical edges; the full topology is maintained to guarantee functional completeness and peak accuracy for all potential tasks.
>
> * **"Pruning" refers exclusively to KV cache optimization, not edge deletion.** Our pruning strategy simply avoids storing the physical KV cache states for low-frequency paths. By prioritizing KV caching for high-frequency paths and falling back to on-the-fly computation for rare ones (Section 5.2), we significantly reduce memory overhead without sacrificing the functional structure of the workflow.
>
> **Q3: Insufficient comparative analysis and positioning against related work**
>
> **A3:** We thank the reviewer for the suggestion to better contextualize our work, and we will expand our related work section in the final manuscript to explicitly contrast GraphFlow with existing literature. While inspired by automated workflow methods like AFlow and AutoFlow, GraphFlow shifts from static template discovery to **dynamic workflow synthesis** over a unified *wGraph* substrate. Furthermore, unlike retrieval-centric frameworks (e.g., AgentKB, TaskWeaver) that treat execution plans as isolated units, GraphFlow explicitly exploits operation-level structural overlap to enable fine-grained procedural recomposition. Finally, compared to general-purpose LLM serving systems (e.g., vLLM, SGLang) that rely on standard prefix caching, GraphFlow introduces **Topology-Aware Differential KV Caching** to efficiently resolve the sparse memory differences across divergent execution paths.
>
> **Q4: Insufficient textual discussion of Figures 1 and 2**
>
> **A4:** We thank the reviewer for this observation. We agree that further discussion of Figures 1 and 2 will enhance clarity and will incorporate an expanded version into the final manuscript.
> - **Figure 1 Discussion:** This figure illustrates a structured agentic workflow for online shopping. It demonstrates how an LLM agent interacts with the environment through discrete atomic operations (such as "Initial Search" and "Deep Filter") and non-linear control-flow primitives, such as conditional decision nodes and iterative loops, to fulfill complex queries.
> - **Figure 2 Discussion:** This figure details GraphFlow’s dual-module architecture. Module 1 depicts the topological reasoning process, using GNN node embeddings and MLP edge scoring to synthesize task-specific subgraphs. Module 2 highlights our topology-aware state management, showing how we store a shared $\mathbf{KV}_{\text{base}}$ and sparse residuals ($\Delta\mathbf{KV}$) to minimize memory redundancy.

---

### Decision · Program_Chairs · 2026-04-30

**Decision:**

Accept (regular)

**Comment:**

This paper introduces GraphFlow, a novel framework that dynamically synthesizes task-specific agent workflows from a unified directed acyclic graph using Graph Neural Networks (GNNs), coupled with a differential KV caching mechanism for memory-efficient serving. The core idea of replacing static workflow templates with dynamic, GNN-driven graph generation is both elegant and highly effective, uniquely bridging high-level agent reasoning with low-level system resource optimization. While one reviewer expressed reservations regarding benchmark difficulty and out-of-distribution generalization, the authors provided a compelling rebuttal featuring additional evaluations on cross-dataset transfers and the interactive ALFWorld benchmark, successfully alleviating these concerns.